# Two Optimization Algorithms for Name-Resolution Server Placement in Information-Centric Networking

**Jiaqi Li** [1,2], **Yiqiang Sheng** [1,2,*] **and Haojiang Deng** [1,2]

1   National Network New Media Engineering Research Center, Institute of Acoustics, Chinese Academy of Sciences No. 21, North Fourth Ring Road, Haidian District, Beijing 100190, China; lijq@dsp.ac.cn (J.L.); denghj@dsp.ac.cn (H.D.)

2   School of Electronic, Electrical and Communication Engineering, University of Chinese Academy of Sciences No. 19(A), Yuquan Road, Shijingshan District, Beijing 100049, China

*   Correspondence: shengyq@dsp.ac.cn; Tel.: +86-1312-116-8320



**Featured Application:** **In the era of 5G (fifth generation)-IoT (Internet of Things) integration, information-centric networking (ICN) is an emerging technology that has the ability to share data on the network layer. It brings great benefits such as in-network caching, mobility support, and inherent security. Our proposed algorithms provide cost-efficient solutions for deploying name-resolution servers in 5G edge networks. They also have the potential to be applied to other scenarios, including the cloudlet placement problem in edge computation and the controller placement problem in software-defined networking.**

**Abstract:** Information-centric networking (ICN) is an emerging network architecture that has the potential to address demands related to transmission latency and reliability in fifth-generation (5G) communication technology and the Internet of Things (IoT). As an essential component of ICN, name resolution provides the capability to translate identifiers into locators. Applications have different demands on name-resolution latency. To meet the demands, deploying name-resolution servers at the edge of the network by dividing it into multilayer overlay networks is effective. Moreover, optimization of the deployment of distributed name-resolution servers in such networks to minimize deployment costs is significant. In this paper, we first study the placement problem of the name-resolution server in ICN. Then, two algorithms called IIT-DOWN and IIT-UP are developed based on the heuristic ideas of inter-layer information transfer (IIT) and server reuse. They transfer server placement information and latency information between adjacent layers from different directions. Finally, experiments are conducted on both simulation networks and a real-world dataset. The experimental results reveal that the proposed algorithms outperform state-of-the-art algorithms such as the latency-aware hierarchical elastic area partitioning (LHP) algorithm in finding more cost-efficient solutions with a shorter execution time.

**Keywords:** 5G; Information-Centric Networking; name resolution; placement optimization; multilayer overlay network

---

## 1. Introduction

As a network technology, the Internet of Things (IoT) [1–3] connects a large number of devices that are integrated with sensing, recognition, processing, communication, and networking functions. Through seamless connections and interactions between a large number of heterogeneous devices, the IoT provides a rich range of services and novel applications, changing the way we live and work [4,5]. In terms of the application area aspect, IoT can be divided into consumer IoT and industrial

IoT (IIoT) [6]. With the rapid development of IoT, there will be 50 billion devices connected to the Internet by 2020, among which about 20% will come from the industrial field, according to Cisco Internet Business Solutions Group predictions [2]. By contrast with consumer IoT scenarios, the IIoT has exceptionally high requirements for transmission latency and reliability. Transmission latency and reliability directly affect the stability of industrial real-time monitoring and automatic control, thus determining the accuracy, efficiency, and costs of industrial production. Not only low average latency but also a deterministic upper bound of latency are needed when handing over data in the IIoT [7]. Due to the increase in the number of IoT devices and the amount of the data they generate, as well as the stringent quality of service requirements of the IIoT, current wireless communication networks, such as 4G, fall short in supporting these challenges, restricting further development of the IIoT.

Fortunately, fifth-generation (5G) wireless communication technology is expected to break the performance bottleneck of the current communication network. Fifth-generation technology has the potential to address the harshest demands posed by the IIoT since it provides higher data rates, higher density, lower end-to-end latency, better reliability, and higher coverage [2,7]. The integration of 5G and IoT will form new connected ecosystems in the near future, becoming one of the major elements in shaping the future Internet [1]. In order to realize the high-performance indicators of 5G, researchers are working hard to improve and standardize radio air interface technology, as well as considering network architecture optimization as an essential part of this [8].

Information-centric networking (ICN) is an emerging network paradigm. It is under standardization and has the potential to enhance and innovate data delivering services in the 5G network [8–10]. ICN adopts the paradigm of separation of the identifier and locator, which shifts the communication model from host-centric to information-centric. It has several characteristics, such as in-network caching [11], mobility support, built-in multicast delivery, and inherent security. These characteristics complement the limitations of semantic overloaded due to exposure to the current Transmission Control Protocol (TCP)/Internet Protocol (IP) network architecture [12,13]. In nature, IoT communications and applications are information-based and follow a content-oriented paradigm. The matching of paradigms also enables ICN to show a good capacity to work with the IoT.

In ICN, the identifier and locator are split into different naming spaces [14,15], and the name-resolution system (NRS) is a network infrastructure that maps and stores the mappings between them. NRS is an essential component of ICN because the delivery of content cannot be realized unless the name-resolution process is completed. Thus, capabilities of name resolution directly influence the performance of ICN in 5G. Several ICN projects have been proposed, and they offer different solutions to the organization and deployment of the NRS. Most of these solutions deploy NRS in the cloud [16], or organize distributed name-resolution servers based on the distributed hash table (DHT) [17–19].

The trend of the future network is transferred from best-effort to deterministic data transmission. Consideration of the upper bound of the name resolution's latency is significant, especially in the 5G-IoT scenario. Emerging technologies such as fog computing [20] and edge computing [21,22] are advancing to enable 5G with ultra-reliable low-latency communication. In [23], Liao et al. introduced the concept of deterministic latency into the NRS and proposed a deterministic latency name resolution (DLNR) framework. The DLNR treats a network as a multilayer overlay network using different latencies because Liao et al. believe that different applications have diverse deterministic latency requirements. For each layer, network entities are partitioned into multiple areas based on the upper bound set by this layer. In addition, name-resolution servers in the DLNR are placed at the edge of the network to achieve lower latency and smaller latency jitter.

Placing servers at the edge of the network is a promising solution that makes servers closer to end-users, providing a short latency response and high rate access [24]. However, this kind of approach implies a substantial increase in deployment and operational costs. Determining how to deploy servers to achieve a tradeoff between users' quality of service and deployment costs is a great challenge.

In this paper, we study how to cost-effectively place name-resolution servers without violating the structural constraints of multilayer overlay networks in the DLNR. As far as we know, most previous research has focused on the server placement problem in a single layer network. Methods proposed in such studies do not work so well in multilayer overlay networks because these methods consider little about the coordination between layers. Thus, it is of great importance to investigate the server placement problem in multilayer overlay networks. The main contributions of this paper are as follows:

- We model the problem of latency-bounded optimal server placement in multilayer overlay networks and formulate it as an integer linear program problem with the objective of minimizing the deployment costs;
- We develop two algorithms based on the heuristic ideas of inter-layer information transfer (IIT) and server reuse. The IIT-DOWN algorithm passes the server placement information from the high-level layer to the low-level layer. It reuses servers chosen in the high-level layer to provide low-level services as well. The IIT-UP algorithm passes the server placement information as well as detailed latency information from low to high. The network scale shrinks during this procedure, and the execution time reduces greatly;
- We conduct experiments on different scales of simulation networks and a real-world dataset to measure the performance of our algorithms. We compare our algorithms with several approaches to solve the server placement problem, and the experimental results show that our algorithms can find more cost-efficient solutions with a shorter execution time.

The remainder of this paper is organized as follows. We review the related work about name-resolution systems and the server placement problem in Section 2. Section 3 presents the system model and problem statement of the name-resolution server placement in multilayer overlay networks. In Section 4, two algorithms are described in detail; then, we evaluate and discuss their performance with a comparison in Section 5. Finally, Section 6 concludes our work.

## 2. Related Work

Two research areas are related to the problem presented in the previous section: name-resolution system and server placement optimization. The following sections present a comprehensive study of the current literature on them.

### 2.1. Name-Resolution System

In existing ICN architectures, name resolution can be mainly divided into two categories according to the coupling relationship with content routing: the name-based routing approach and the standalone name-resolution approach [12,25,26]. We do not focus on name-based routing approaches, such as Content-Centric Network (CCN) [27] and Named-Data Network (NDN) [28], because name resolution and content routing are coupled in this approach, and there is no need for a separate name-resolution service. ICN architectures using standalone name resolution include Data-Oriented Network Architecture (DONA) [29], Publish/Subscribe for Internet Routing Paradigm (PSIRP) [30], MobilityFirst [16], and so on. In this approach, name resolution and content routing are decoupled into two steps: the identifier is used to lookup associated locators, and then data is routed by the locators.

Organization and deployment of NRS are crucial in standalone name-resolution approaches, as they directly affect the efficiency of name resolution and the subsequent routing. MobilityFirst employs a method called Globel Name Resolution Service (GNRS) with realization in the cloud, which leads to long resolution latency [16]. Aiming to enhance scalability, several DHT-based schemas have been proposed and studied, such as Multi-level Distributed Hash Table (MDHT) [17], Hierarchical Distributed Hash Table (HDHT) [18], Scalable Multi-level Virtual Distributed Hash Table (SVMDHT) [19], and Hierarchical Pastry (H-Pastry) [31]. Despite achieving good scalability and robustness, the DHT mechanism has an inherent disadvantage that it is not topology aware. Overlay networks built by DHT do not consider the nearness between nodes, and the name-resolution latency

is hard to achieve. Compared with DHT methods, tree-based methods have a more definite query path while taking scalability into account. Sun et al. proposed Griffin, which is based on a tree structure and provides an efficient and scalable name-resolution service. However, its structure is built statically and it cannot adapt to the dynamic environment in the 5G-IoT scenario [32]. Ftree, developed by Louati et al. [33], stores name mapping in leaf nodes to ensure the nearest response is achieved, but it does not distinguish among different latency demands, and its name-resolution latency still does not promise an upper bound.

## 2.2. Deterministic Latency Name Resolution

Designing an ICN name-resolution system that can effectively meet the high scalability and deterministic latency requirements of 5G-IoT is extremely challenging. Figure 1 shows the DLNR framework. It is a potential solution proposed by Liao et al. [23]. DLNR develops a locally enhanced name-resolution mechanism to provide a one-to-many relationship between an identifier and locators with the constraints of scopes or distances to achieve deterministic low latencies in a limited domain by accelerating the name-resolution process. It supports flat-name based NRS schemas, and the distributed name solution nodes in this schema are organized by a tree structure. By contrast with current NRS approaches, DLNR focuses on the underlying network and partitions the network entities into several nested hierarchical elastic areas (HEAs) according to their different deterministic latency constraints. There is a dedicated name-resolution node called the HEA Manager (HM) in every HEA. The HM is responsible for establishing, storing the mapping of identifiers to locators and it provides the primary name-resolution service to requesters in the HEA. In DLNR, the physical network is partitioned into multilayer overlay networks, corresponding to multiple level HEAs. A higher-level HEA consists of several lower-level HEAs. Different level layers of HEA have different latency constraints. The higher level means a broader coverage and a higher latency upper bound. HEAs in the same level hold the same latency with no area overlap. It is worth noting that Liao et al. carefully analyzed the proportion of latencies contributed by each procedure in the name-resolution event. They found that the transmission latency was much greater than that of the other steps. Therefore, they used the transmission latency as the organization basis of the system structure, which is also used in this paper.

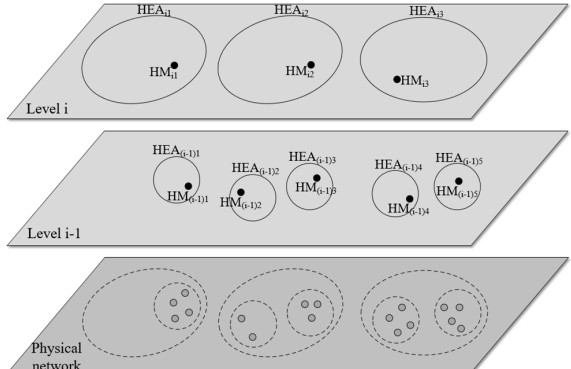

**Figure 1.** The nested structure of deterministic latency name resolution (DLNR). The physical network is partitioned into several hierarchical elastic areas (HEAs). These HEAs are nested organized, and each HEA has an HEA Manager (HM) to provide a name-resolution service. Each layer is constrained by an upper bound of latency.

The design of DLNR effectively addresses the 5G-IoT requirements. The method of the hierarchical organization improves the scalability and satisfies differentiated demands for latency. The partitioning of HEAs realizes the name resolution of deterministic latency, which limits the queries in the domain and reduces the query traffic between domains. The nested tree structure improves the management efficiency. The area partitioned according to the latency can be constructed elastically. The networks

in DLNR are typical multilayer overlay networks, and a reasonable partition for each layer and the placement of name-resolution servers are the keys to the whole architecture working efficiently.

*2.3. Name-Resolution Server Placement in Information-Centric Networking (ICN)*

As mentioned above, DLNR is an organizational framework for ICN name-resolution services in 5G-IoT scenarios. It brings two major constraints to the name-resolution server placement problem. Firstly, the name-resolution server is compulsive at providing a resolution service within an upper bound of latency. Secondly, name-resolution servers are organized as a multilayer to satisfy different requirements of name-resolution latency. Proper placement of the servers is critical to reduce the costs of system deployment. Therefore, we undertook a lot of research for inspiration.

Proper placement of the servers is critical to reduce the costs of system deployment in multilayer overlay networks, such as the DLNR mentioned above. The server placement problem is a typical problem that is widely applied in cloudlet placement [34], controller deployment in software-defined networking [35], and virtual machine placement in cloud computing [36]. However, the server placement problem can usually be regarded as an NP-hard problem [37], so it has attracted the attention of researchers seeking to develop efficient algorithms to reach an approximate solution within a reasonable amount of time.

Jia et al. [38] studied how to place K cloudlets and allocate users to cloudlets in wireless metropolitan area networks (WMAN) such that the average system response time is minimized. They proposed a density-based clustering (DBC) algorithm to solve the problem with the aim of enabling the placement of cloudlets at regions with high user density. Xu et al. [39] formulated a novel capacitated cloudlet placement problem that placed K cloudlets in different strategic locations. They showed that the problem is NP-hard and then devised a greedy-based heuristic algorithm to solve it. Since the greedy-based algorithm is simple and easy to implement to solve this problem, several k-means based algorithms have been presented [40,41]. Wang et al. [42] defined an edge-server problem as a multi-objective constraint optimization problem. They proposed an exact mixed-integer linear programming solution to minimize the total access delay between users and edge servers. Moreover, several heuristic algorithms were proposed to solve the server placement problem. In [43], Jia et al. proposed a fast heuristic and a distributed genetic algorithm to minimize the maximum response time of all offloaded tasks. The authors in [44] proposed a new heuristic algorithm and a particle swarm optimization algorithm to find better server placement solutions.

The studies mentioned above focused on the placement strategy with K servers. Their targets were to minimize the average access time and balance of offloading rather than minimizing the costs of servers. In [45], Ma et al. placed cloudlets in the WMAN using a clustering algorithm, K-Medoids, with the objective of minimizing the count of cloudlets. Li and Wang formulated the problem as a multi-objective optimization problem and devised a particle swarm optimization algorithm to find the optimal solution [46]. Studies in [47,48] transformed the edge-server placement problem into an optimization problem that requires the dominating set of a given graph to be minimized, and graph algorithms were applied to solve the problem.

Most of the studies on placement problems have focused on single-layer networks. When it comes to multilayer networks, the problem becomes even more complicated, because the simple superposition of optimal strategies in a single layer usually cannot obtain the optimal strategy in multi-layer networks. Liu et al. [49] studied the hierarchical deployment of mobile edge computing servers and the user allocation problem. Their hierarchical deployment is relative to specific 5G entities, and no upper limit to access delay was considered. In [50], Sinky et al. studied placement problems in tree-like structures. They used a hierarchical clustering algorithm with good scalability that only adapts to the K server placement problem.

The studies above have some limitations if directly applied, because they do not take into account both deterministic latency constraint and multiple latency demands when name-resolution servers are placed in ICN. The most relevant studies are [23,51]. In [51], Nacher et al. first defined the multilayer

control problem in terms of the minimum dominating set (MDS) controllability framework. Their target was to minimize the total dominator count, which is similar to the server placement problem. However, there are no more constraints between layers in their model, and it is not suitable for direct application in name-resolution server placement in our situation. Liao et al. [23] proposed the DLNR framework, and an algorithm called the latency-aware hierarchical elastic area partitioning (LHP) algorithm was also proposed. However, the LHP is just a feasible algorithm that outputs a placement and partition solution to satisfy the structural constraints, and its costs are not minimized.

## 3. System Model and Problem Statement

We modeled the placement problem of name-resolution servers in multilayer overlay networks with an undirected graph $G(V, E, W)$. $V = \{v_1, v_2, \ldots, v_N\}$ is the set of nodes in the physical network, including all network entities with the potential to place name-resolution servers. $N$ is the total count of network entities. $E$ is the set of edges, and edge $e(u, v)$ belongs to $E$ if node $u$ and node $v$ can communicate with each other without going through any other nodes in $V$. $W$ is the weight set of $E$, and the weight $w(u, v)$ is the one-way transmission latency between node $u$ and node $v$. We used a matrix $D$ to denote the one-way transmission latencies for all pairs of nodes in $G$, where $d(u, v)$ represents the shortest path latency between node $u$ and node $v$. $T = \{t_1, t_2, \ldots, t_L\}$ is a set that consists of upper bounds of each layer's constraint latencies, and $L$ is the count of different latency scenarios. It is noteworthy that when it comes to the secure name-resolution scenarios, authentication and encryption processes may take some time. In these scenarios, upper bounds in $T$ should be calculated by subtracting corresponding process time. For all $0 \leq i < j \leq L$, $t_i < t_j$ holds, which means that higher-level HEAs have higher latency upper bounds. $T$ is usually determined by application scenarios in the network. The detailed notations and descriptions used in this paper are summarized in Table 1.

**Table 1.** Summary of notation.

| Notation | Description |
|:---:|:---:|
| $V$ | the set of nodes that have the potential to place servers |
| $N$ | the total count of nodes in $V$ |
| $E$ | the set of links directly connect between nodes |
| $W$ | the set of one-way transmission latencies of links |
| $D$ | $N \times N$ matrix, the shortest path latencies between every pair of nodes in $V$ |
| $e(u, v)$ | the link between node $u$ and node $v$ |
| $w(u, v)$ | the one-way transmission latencies of link $e(u, v)$ |
| $d(u, v)$ | the shortest path latency between node $u$ and node $v$ |
| $T$ | the set of upper bounds of each level layer's constraint latencies |
| $L$ | the count of layer levels |
| $i$ | the index of a level |
| $M^i$ | the count of HEAs in level $i$ |
| $m$ | the index of HEA |
| $HEA_m^i$ | the m-th cluster in level $i$ |
| $HM_m^i$ | the cluster head of the m-th HEA in level $i$ |
| $H_m^i$ | the cluster member set of the m-th HEA in level $i$ |
| $x_v^i$ | binary variable, equal to 1 if node $v$ is chosen as an HM in level $i$ |
| $y_v^i$ | binary variable, equal to 0 if node $v$ is chosen as HM in the level higher than $i$ |
| $\alpha^i$ | the costs of deploying a server at level $i$ |
| $S_{HM}$ | the global set of HM |
| $ML^i$ | the attribute of a HM to record the maximum latency in its HEA |

In this model, we can describe our optimization objective and the constraints of multilayer overlay networks more accurately with exact formulas. For a layer in level $i$ with a latency upper bound of $t_i$, we assume that $G$ is divided into several HEAs. We record the count of HEAs as $M^i$. Nodes and edges in each HEA form a subgraph $G_m^i(V_m^i, E_m^i, W_m^i)$ of $G$, $m = 1, 2, \ldots, M^i$. $HEA_m^i$ represents the cluster of $G_m^i$, and $HM_m^i$ represents the cluster head of $HEA_m^i$. For all nodes $v$ in $V$, if node $v$ is chosen as $HM_m^i$, we record $x_v^i = 1$, otherwise, $x_v^i = 0$. In our model, different overlay layers can choose the same physical node for server placement, so we let the costs of deployment of the highest level represent the

costs of this physical server. If node $v$ has been chosen as an HM in the level higher than $i$, we record $y_v^i = 0$, otherwise, $y_v^i = 1$.

Let $\alpha^i$ represent the cost of deploying a server at level $i$. The server placement problem in multilayer overlay networks can be formulated as follows:

Minimize

$$Cost = \sum_{i=1}^{L} \sum_{v \in V} x_v^i * y_v^i * \alpha^i, \tag{1}$$

subject to

$$\cup_{m=1}^{M^i} V_m^i = V, \quad \forall i \in \{1, 2, \ldots, L\}, \tag{2}$$

$$V_m^i \cap V_n^i = \varnothing, \, \forall m, n \in \left\{1, 2, \ldots, M^i\right\}, \, m \neq n, \quad i \in \{1, 2, \ldots, L\}, \tag{3}$$

$$V_m^i \subseteq V_n^j, \, if \, \exists v \in V_m^i \, and \, v \in V_n^j, \, m \in \left\{1, 2, \ldots, M^i\right\}, \quad n \in \left\{1, 2, \ldots, M^j\right\}, \tag{4}$$
$$0 \leq i < j \leq L,$$

$$d\left(v, HM_m^i\right) < t_i, \, \forall v \in V_m^i, \, m \in \left\{1, 2, \ldots, M^i\right\}, \, \forall \, t_i \in T. \tag{5}$$

Equation (1) represents the total placement costs of name-resolution servers. The constraint of Equation (2) guarantees the full coverage of the partitioned HEAs to the nodes in the network, which means all requesters can use name-resolution services in each latency level. The constraint of Equation (3) ensures that there is no overlap between HEAs in the same layer, so a tree structure can be formed and the management work becomes easy. Nested relationships between HEAs from different level layers are indicated in Expression (4). In Expression (5), the upper bounds of name-resolution response latencies are promised.

If a solution of placing name-resolution servers is subject to the aforementioned constraints, it is a feasible solution in server placement in multilayer overlay networks. Finding the HM set of a single layer network with a minimum count is similar to the MDS problem in graph theory. Nacher et al. [51] first defined the multilayer control problem in terms of the MDS controllability framework, which is called MDS in multilayer. They demonstrated that even in special cases of networks in which the MDS is solved in polynomial time, the MDS in a multilayer (MDSM) problem is still NP-hard. There are several constraints to guarantee the structure characteristic of multilayer overlay networks in our model, and the server placement problem is NP-hard as well. A good algorithm gives a solution that outputs HEAs and HMs for every layer of $G$ with minimum cost and a shorter execution time.

## 4. Proposed Algorithms

In this section, we analyze the characteristics of the server placement problem. Inspired by the ideas of transfer placement information between layers and server reuse, we developed the IIT-DOWN algorithm in the direction from the high-level layer to the low-level layer. Then, we found that the servers in the lower-level layer occupy the majority of the total number of servers, so we proposed the IIT-UP algorithm, which transfers placement information and detailed latency information from low to high. In these two algorithms, we used the relaxation technique of binary integer linear programming in choosing each layer's HM, since it is an NP-hard problem as well. Though the result is approximate, it can output a solution that is not far from the exact solution within polynomial time [52].

### 4.1. Inter-Layer Information Transfer (IIT)-DOWN Algorithm

As shown in Figure 2, the main idea of the IIT-DOWN algorithm is as follows. After choosing the HMs from a certain layer network, IIT-DOWN passes the chosen HMs' location information to the lower-level layer and removes the nodes in these locations. The details of this algorithm are as follows: We carry out HEA partitioning from a high latency level to a low latency level. First, we deal with the highest-level layer. All nodes are painted white in $G$, which means that these nodes are not members of any HEAs yet. Then, the edges with a latency higher than $t_i$, the latency upper bound of this level, are removed. On the other hand, we add an edge between node $u$ and node $v$ with

$w(u, v) = d(u, v)$, if $d(u, v) < t_i$ holds. Then, we calculate the current layer's HMs with the minimum number and add these nodes to the global HM set, marked as $S_{HM}$, and all nodes in $S_{HM}$ are painted black. For every node in white, we find its closest HM from $S_{HM}$, and make this node a member of the corresponding HEA. The nodes contained in the HEA are marked as H. Then, we undertake lower-level HEA partitioning for every partitioned higher-level HEA. In a lower-level layer, we remove and add edges with the same principle. Then, we remove the nodes that are chosen as HMs in the level higher than the current level and paint their neighboring nodes grey. By now, the current layer's HMs can be calculated using algorithm 3. Iteratively, after finishing partitioning in every layer, the output is the partitioning result containing HEAs and $S_{HM}$. The specific algorithm is presented in Algorithm 1.

---

**Algorithm 1: IIT-DOWN**

---

**Input:** $G(V, E, W), D$
**Output:** $S_{HM}, Hs$
**Parameters:** $T = \{t_1, t_2, \ldots, t_L\}$
1:  Initialization: $H^{L+1} = \{V\}, HM^{L+1} = \varnothing$
2:  for $i$ from $L$ to 1 do
3:      for each $h$ in $H^{i+1}$ do
4:          generate $G^i\left(V^i, E^i, W^i\right)$ where $V^i = h, E^i = \varnothing, W^i = \varnothing$
5:          for $v$ in $V^i$ do
6:              $v$.color = white
7:              if $v$ in $HM^{i+1}$ then $v$.color = black
8:          for each $u, v$ in $V^i$ that $u \neq v$ do
9:              if $d(u, v) < t_i$ then
10:                 add $e(u, v)$ to $E^i$ and add $w(u, v) = d(u, v)$ to $W^i$
11:         for $v$ in $HM^{i+1}$ do
12:             for $u$ in $V^i$ do
13:                 if $e(u, v)$ in $E^i$ and $u$.color = white then $u$.color = grey
14:             remove $v$ from $G^i$
15:         $HM^i$ = Choose_HM $(G^i)$
16:         for each $v$ in $V^i$ do
17:             if $v$ in $HM^i$ then $v$.color = black
18:             if not $v$.color = black then
19:                 find $k$ with the minimum $d(v, k)$ in $HM^i$ and add $v$ to $H_k^i$
20:         add $H_k^i$ to $H^i$
21:         add $HM^i$ to $S_{HM}$ and add $H^i$ to $Hs$
22: return $S_{HM}, Hs$

---

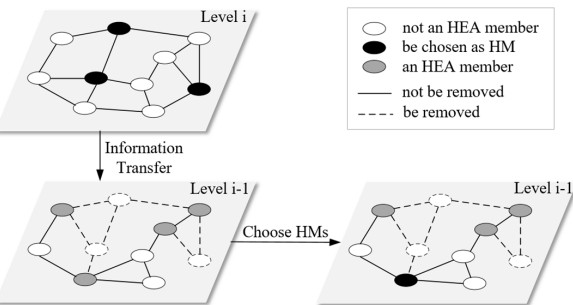

**Figure 2.** An example operation in inter-layer information transfer (IIT)-DOWN. Nodes do not change in different layers, but the connections among them may be different. Nodes in black are chosen for the placement of name-resolution servers. Nodes in white represent those that have not been partitioned. The HMs chosen from level $i$ are removed in level $i - 1$, and grey nodes represent nodes that satisfy the latency constraint from these HMs. There are four HMs in total in this example.

## 4.2. IIT-UP Algorithm

While studying characteristics of multilayer overlay networks, we found that the topology of the lower-level layer is a subgraph of the higher-level layer network. Because the level is determined by a different latency, once two nodes can communicate with each other in a low-latency level, they can communicate no more than a high-latency upper bound. On the other hand, we found that it is servers in the lower-level layer that occupy the majority of the total count of servers. We took full advantage of these characteristics and proposed the IIT-UP algorithm. This is an algorithm that gives priority to the processing of the lower-level layer network. The topology that contains HMs in the sublayer forms a subgraph of the physical network and becomes the current layer's input.

As shown in Figure 3, when dealing with the layer of level *i*, edges in which latency is higher than the upper bound of this level are deleted, and each pair of nodes is connected if the transmission latency between them is lower than the upper bound. Then, the current layer's HMs are calculated with the minimum number using the relaxation technique of binary integer linear programming. For every node that is not in an HM, its closest HM is found and this node is made to be a member of the corresponding HEA. After this procedure, every HM has several HEA members. The maximum latency among the latencies of HEA members to their HM is found and recorded as an attribute of this HM, marked $ML^i$ (maximum latency). Then, partitioning in the level $i + 1$ layer begins, where the nodes of the level $i + 1$ layer come from $HM^i$. Accordingly, the principles of addition and deletion edges for every node are constrained by the difference between $t_i$ and $ML_v^i$. Iteratively, after finishing partitioning in every layer, the output is the partitioning result containing HEAs and $S_{HM}$. The specific algorithm is presented in Algorithm 2.

---

**Algorithm 2: IIT-UP**

---

**Input:** $G(V, E, W), D$
**Output:** $S_{HM}, Hs$
**Parameters:** $T = \{t_1, t_2, \ldots, t_L\}$
1:   Initialization: $HM^0 = V$
2:   for each $v$ in $V$ do
3:      $v.ML^i = 0$
4:   for $i$ from 1 to $L$ do
5:      generate $G^i\left(V^i, E^i, W^i\right)$ where $V^i = HM^{i-1}, E^i = \varnothing, W^i = \varnothing$
6:      for each $v$ in $V^i$ do
7:        $v.color$ = white
8:      for each $u, v$ in $V^i$ that $u \neq v$ do
9:        if $d(u, v) < t_i - \max\left(u.ML^i, v.ML^i\right)$ then
10:          add $e(u, v)$ to $E^i$ and add $w(u, v) = d(u, v)$ to $W^i$
11:      $HM^i$= Choose_HM $\left(G^i\right)$
12:      for each $v$ in $V^i$ do
13:        if $v$ in $HM^i$ then $v.color$ = black
14:        if not $v.color$ = black then
15:          find $k$ with the minimum $d(v, k)$ in $HM^i$ and add $v$ to $H_k^i$
16:      add $H_k^i$ to $H^i$
17:      add $HM^i$ to $S_{HM}$ and add $H^i$ to $Hs$
18:   return $S_{HM}, Hs$

---

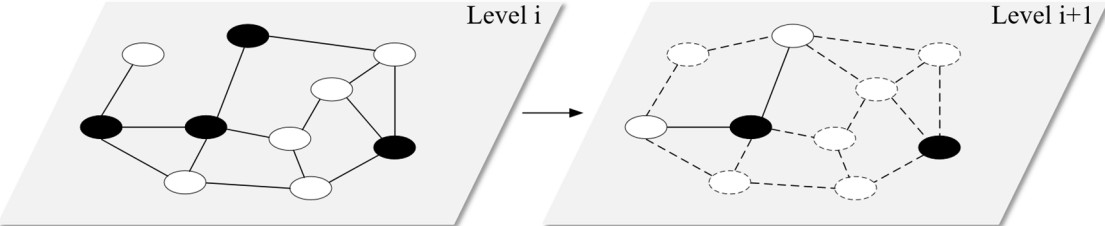

**Figure 3.** An example operation in IIT-UP. The network of each layer is solved in order from low to high. The HMs in the level *i* layer constitute the topology of the level $i + 1$ layer.

### 4.3. Computation of Hierarchical Elastic Areas Manager (HM) in a Single Layer

In IIT-DOWN and IIT-UP, a solution for a single layer network is needed as well. This problem is similar to finding an MDS set of a network. Here, we use the relaxation technique of binary integer linear programming to choose the HMs in the colored graph in the proposed algorithms. The specific algorithm is presented in Algorithm 3. However, other efficient single-layer approximation algorithms can also be applied here, but this is not the focus of the current research.

---

**Algorithm 3: Choose_HM**

---

**Input:** $G(V, E, W)$
**Output:** *HMs*
1:  Initialization: *Vars* = ∅
2:  for each $v$ in $V$ do
3:    add a variable $x_v$ to *Vars*, $x_v \in [0, 1]$
4:  for each $v$ in $V$ do
5:    if $v$.color = white then
6:      add constraint $x_v + \sum_{u:\, e(u,\ v) \in E} x_u \geq 1$
7:  minimize $\sum_{v:\, v \in V} x_v$ as optimize objective
8:  do optimize
9:  for each $x_v$ in *Vars* do
10:    if $x_v > 0$ then add $v$ to *HMs*
11:  return *HMs*

---

## 5. Evaluation and Discussion

### 5.1. Simulation Network

For comparison's sake, we reproduced several existing algorithms and applied them to the placement problem of name resolution in multilayer overlay networks. The comparison algorithms were as follows:

- LHP: this algorithm was described in [23], and it is a heuristic graph-partitioning algorithm. It divides a physical network into one or more connected subgraphs with the latency level constrained from high to low. It chooses name-resolution nodes with the maximum degree and finds nodes they cover. Partitioning for each layer is achieved one by one;
- MDSM: this is the algorithm that was used to analyze the upper bound of MDS in multilayer networks in [51]. It is an algorithm that removes the neighboring nodes of higher-level dominators before solving the current level layer's MDS. The original MDSM does not satisfy the latency constraints in multilayer overlay networks. We extended the MDSM appropriately to adapt to the problem;
- Random allocate (RA): in this algorithm, the HM is chosen randomly, and users are allocated to an HEA, depending on which HM is closest to them.

We generated random graphs that follow the Barabási-Albert (BA) scale-free model. It is a typical power-law degree distribution network that is widely used to simulate real network topology [53]. These networks' node sizes are from 100 to 2000, and the average degree is 2. According to [54], the end-to-end latency of the current internet is in the magnitude of ten milliseconds. Thus, we randomly set the weight of each edge from 0 to 10 ms. Considering the different latency requirements of major application scenarios in 5G-IoT, we chose $T = \{10\ ms, 25\ ms, 50\ ms\}$ as the upper bounds of the one-way transmission latency [55]. These bounds cannot represent all of the latency demands but are already able to evaluate the differences among these algorithms objectively. When it came to solving the HM choosing problem in a single layer, we used the Solving Constraint Integer Programs (SCIP) solver. It is currently one of the fastest non-commercial solvers for mixed integer programming and mixed integer non-linear programming [56]. The settings of the experiment are also shown in Table 2.

**Table 2.** Evaluation setting.

| Parameter | Value |
| --- | --- |
| Number of nodes | 100~2000 |
| Average degree | 2 |
| $L$ | 3 |
| $T$ (ms) | 10, 25, 50 |
| Latency scope (ms) | (0, 10] |

Figure 4 shows the network scenario of simulation experiments. The network topology consists of edge nodes such as base station and access point. Mobile edge computing in 5G technology makes it possible to deploy name-resolution servers at the edge of the network. The expected server placement result is also shown in the figure. The network is divided into three layers, corresponding to different upper bounds of name-resolution latency. At each layer, several edge nodes are selected to place name-resolution servers to provide deterministic resolution services.

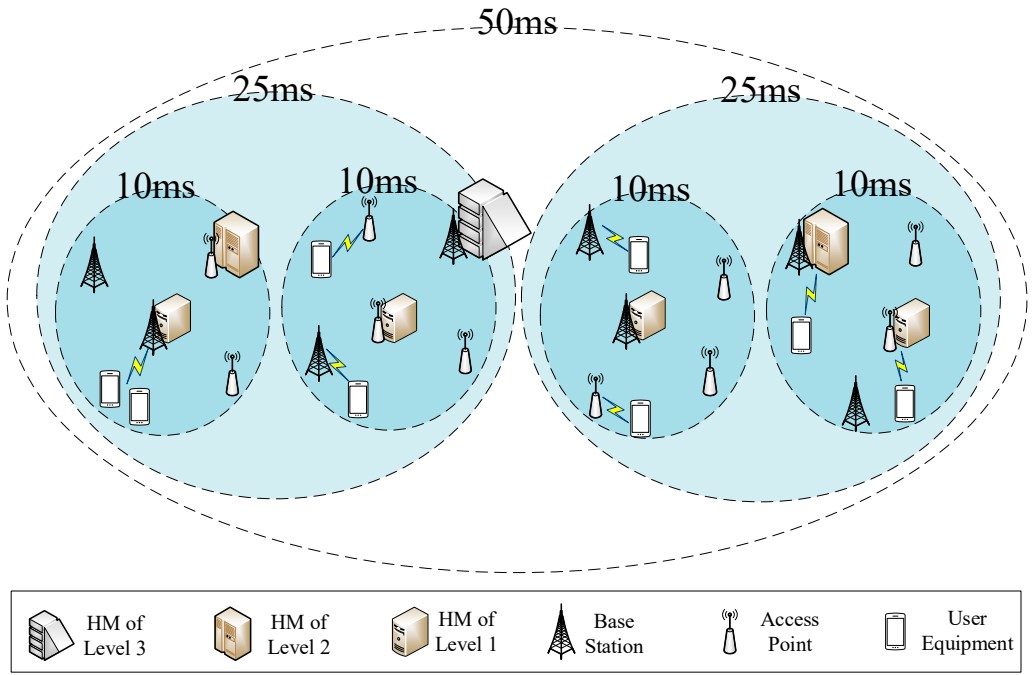

**Figure 4.** The network scenario of simulation experiments.

The experiment environment was created and run in Python 3.6 on a PC with an Intel Core (TM) i7-9750H CPU and 16GB of RAM. For each network node size, we experimented with 15 trials.

All algorithms were run with the same experiment environment and input topology. We recorded several indicators for analysis, including the costs, execution time, number of HMs, and average latency. The performances we analyzed were as follows.

### 5.1.1. Deployment Costs

We assumed that the costs of each server were the same and calculated the total costs of servers that were placed to satisfy the transmission latency constraints under different network scales using each algorithm. The simulation results are shown in Figure 5. It can be seen that the two IIT algorithms performed better than the LHP algorithm and the RA algorithm in finding cost-efficient solutions. The performance of the MDSM algorithm was in the middle. We analyzed the data from the experiment. The cost of the solution obtained by IIT-DOWN was about 59.6% of that obtained by the LHP algorithm, and about 67.9% of that obtained by MDSM. For the IIT-UP algorithm, the ratios of the same comparison were 56.8% and 64.8%, respectively. We can draw the conclusion that the costs of deployment can be effectively reduced by passing information between layers to coordinate server placement and reusing servers. In addition, the relationship between costs and network size was shown to be linear for all algorithms.

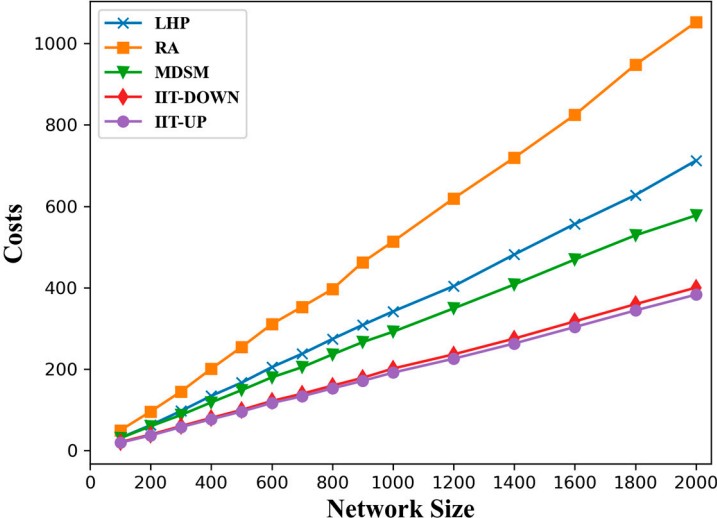

**Figure 5.** Deployment cost comparison for each algorithm with different network sizes.

### 5.1.2. HM Count

We also analyzed the number of HMs in each layer, as shown in Figure 6. We figured out that IIT algorithms can reduce the number of HMs at each level. Although the IIT-UP algorithm gives the best results in terms of cost, we found that the IIT-DOWN algorithm outputs a lower HM count in the higher-level layers, because it gives priority to dealing with the higher-level layer. In the situation where servers deployed at the higher-level layer use more resources, such as bandwidth, storage, and computation, IIT-DOWN becomes the best choice.

We mentioned above that the HM choice problem is similar to the MDS problem in graph theory, Nacher et al. proved that in the BA scale-free model, the size of the MDS is $\theta(n)$ [57], where n is the number of nodes in the network. Our simulation results also confirmed this conclusion. We can use this conclusion to estimate the number of name-resolution servers we have to deploy in a target network if the network is in line with the BA scale-free model.

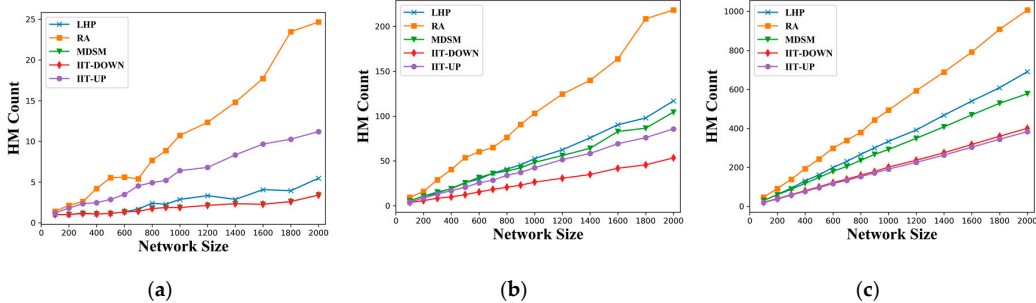

**Figure 6.** HM count at each layer. In each layer, inter-layer information transfer (IIT) algorithms perform better than other algorithms. IIT-DOWN and multilayer minimum dominating set (MDSM) algorithms need less name-resolution servers in higher-level layers. (**a**) The layer with the latency upper bound at 50 ms. Note that the results of IIT-DOWN and MDSM overlap. (**b**) The layer with the latency upper bound at 25 ms. (**c**) The layer with the latency upper bound at 10 ms.

### 5.1.3. Execution Time

With an increment in the network size, the network becomes more and more complex. The time to solve the problem will become increasingly more difficult to accept. When minimizing the costs of server deployment is considered the optimization goal, we must consider the operation efficiency of the algorithm. As shown in Figure 7, we compared the execution time of each algorithm on different network scales. First, we figured out that the execution times of these algorithms increased sharply with an increment in the network size. Then we found that IIT algorithms output better results in less time compared with the LHP algorithm. The MDSM algorithm removes neighbor nodes to reduce the complexity and accelerate problem resolving. Still, we can see that the running times between it and IIT-DOWN have little difference. Among these algorithms, IIT-UP has the best operation efficiency, which means it can handle more extensive networks within a reasonable solving time.

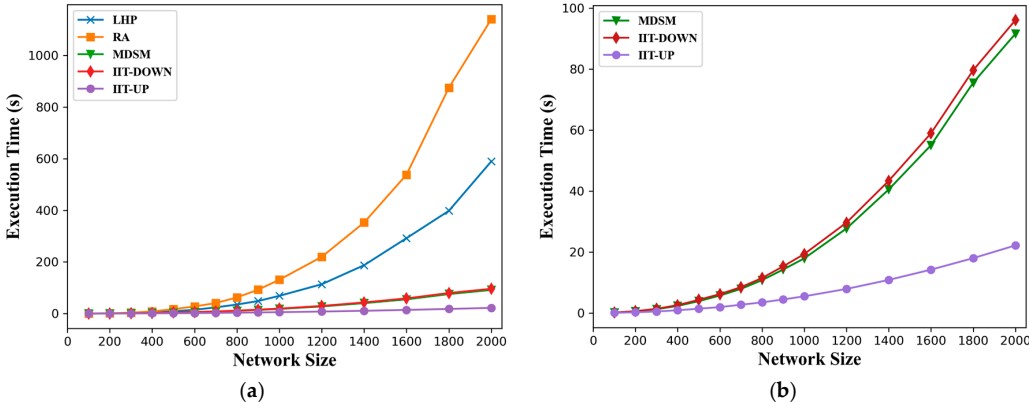

**Figure 7.** Execution time comparison of each algorithm with different network sizes. IIT algorithms require less time to solve a problem, which means that they can solve problems in more extensive networks. The latency-aware hierarchical elastic area partitioning (LHP) algorithms may not complete the calculation in a reasonable time. (**a**) Comparison of all five algorithms. (**b**) Comparisons of MDSM, IIT-DOWN, and IIT-UP are presented in detail.

### 5.1.4. Average Latency

In multilayer overlay networks, it is enough for us to satisfy the requirement that the data transmission time does not exceed the upper bound of latency. We also analyzed the average latency at each level after the HEA partition using these algorithms. Figure 8c shows that at the bottom layer with the upper bound latency of 10 ms, LHP algorithms result in a lower average latency, and the algorithms proposed by us are not as good. However, it is unfair to only compare the average latency because the

LHP algorithm uses more servers than IIT algorithms. Interestingly, when it comes to higher-level layers (Figure 8a,b), IIT algorithms result in a lower average latency, although we know that the LHP still uses more servers. This may be caused by the user allocations strategy and the accuracy of the HEA partition. The IIT-UP algorithm has the shortest average latency in higher-level layers, which means it has the most accurate partition and the highest server utilization. Also, we found that the average latency does not increase with an increment in the network size, mainly because the number of HMs grows linearly with the size of the network, and the balanced partition maintains the stability of the average latency.

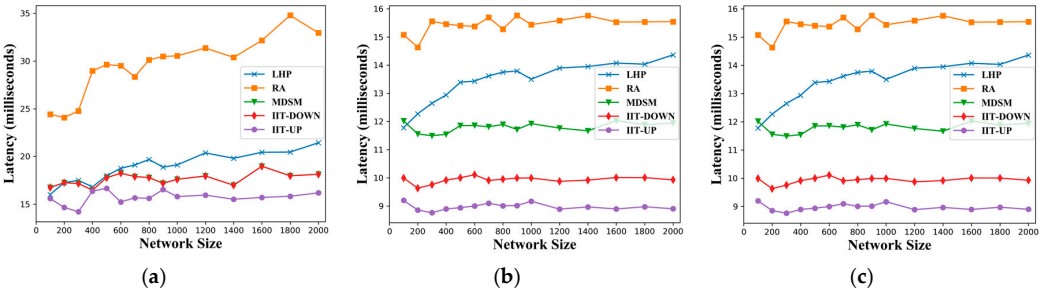

(**a**)          (**b**)          (**c**)

**Figure 8.** Average name-resolution latency at each layer. LHP provides a shorter average latency at the lowest level because it uses more name-resolution nodes. However, in higher-level layers, IIT algorithms perform better. (**a**) The layer with the latency upper bound at 50 ms. Note that the results of IIT-DOWN and MDSM overlap. (**b**) The layer with the latency upper bound at 25 ms. (**c**) The layer with the latency upper bound at 10 ms.

### 5.1.5. Cost Parameter

In Section 5.1.1, name-resolution servers at different latency levels are assumed to have the same deployment cost weight. However, in reality, servers that cover a broader range often require higher configurations, such as central processing unit (CPU), bandwidth, and storage, which also incurs higher costs. Assuming that the ratio of deployment costs for servers at adjacent latency layers is $k$, a further study about the impact of $k$ on deployment costs was undertaken. Figure 9 shows the change in the total cost relative to the $k$ value under the three algorithms when the network size is 1000. When the difference in server costs at different levels is small, IIT-UP gives the solution with the lowest cost, since it can minimize the total number of servers. However, as $k$ increases, its cost will gradually exceed that of IIT-DOWN and MDSM. The exceedance points were 1.45 and 5.9 in this experiment, respectively. In addition, we noticed that the IIT-DOWN algorithm always has a cost advantage over the MDSM approach.

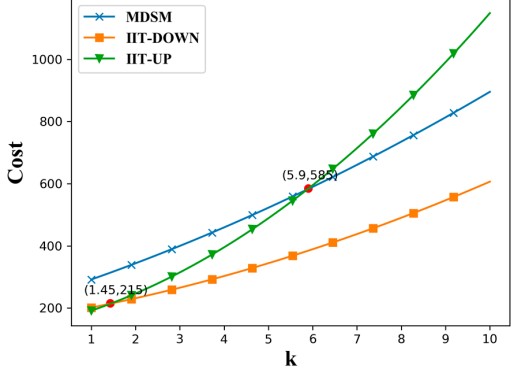

**Figure 9.** Impact of $k$ on deployment costs.

### 5.2. Coverage in K Placement Algorithms

The study above was undertaken to minimize the server cost under the constraint of the deterministic delay structure. The count of servers cannot be forecast before solving the problem. When the budget is limited, the server placement problem turns into the K placement problem. We also set up experiments to find proper methods that can be used in this situation. We compared our proposed IIT-UP algorithm with several algorithms, as follows:

- Density-based clustering (DBC) [38]: DBC first selects one node which has the largest degree among all the nodes in the network and places one server on this node. Subsequently, all users that have an access delay within this server's coverage and have not been allocated are allocated to this server. This procedure continues until all K servers are placed in the network, or all users are allocated;
- K-Mediods [45]: this approach is a variant of K-means, which is commonly used to cluster a data set into K groups automatically. In this approach, K initial cluster centers are selected and then iteratively refined. In every iteration, a new cluster center is selected to minimize the within-cluster sum of the access delay. This procedure continues until no cluster center changes, and the last iteration's cluster centers are placed servers;
- Top-K: this approach places the K servers on k nodes with the largest degree. Users are allocated to their closest servers;
- Random-K: this approach randomly selects K nodes and places servers on them. Users are allocated to their closest servers.

Since the approaches above mainly focus on single-layer networks, we extended them accordingly to adapt them to the situation of multilayer overlay networks with nested constraints. The IIT-UP algorithm was set as a benchmark. That is to say, in every layer, the parameter K was chosen from the results output by IIT-UP. The experiment also involved 15 turns, and the same topologies were guaranteed for each approach in each turn. The determined-access-latency coverage rate of users was used as an evaluation of the performance of each approach. As shown in Figure 10, the IIT-UP algorithms always guarantee the full coverage rate, because the strict latency constraint is applied in the procedure of server selection and user allocation. However, uncovered users exist in other approaches. The primary trend that we figured out is that as the level goes from high to low, the coverage rate becomes lower. Among these approaches, DBC outperforms other K placement approaches and can maintain more than 90 percent of the coverage in each layer. The K-Mediods approach performs at a slightly lower level. An impressive result in our experiment was that in level 1 (Figure 10c) and level 2 (Figure 10b), the random-K approach even outperformed the Top-K approach. This means that if we simply choose top-k degree nodes for server placement, this may result in more unsatisfactory results.

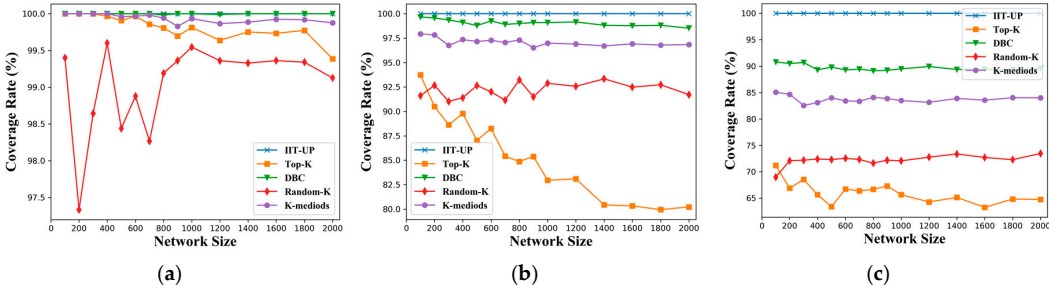

**Figure 10.** The coverage rate of K placement algorithms. (**a**) The layer with the latency upper bound at 50 ms. (**b**) The layer with the latency upper bound at 25 ms. (**c**) The layer with the latency upper bound at 10 ms.

### 5.3. Real-World Dataset

We also utilized the dataset for Shanghai Telecom's base stations [58,59] for algorithm comparison. Shanghai is a typical densely populated city, and its base station distribution is also dense. This dataset contains location information for more than 3000 base stations. In our experiment, we randomly chose 1000 base stations and used their location information to calculate the Euclidean distances between each pair of base stations. Then, we calculated the propagation latency between two base stations by considering an approximate propagation time of 5 μs/km [60]. Since the spans of distances among Shanghai base stations are small, we chose T = {0.01 ms, 0.05 ms, 0.2 ms} as the corresponding narrowing.

Table 3 shows the results of each algorithm, including the execution time, name-resolution server at each layer, and total cost. The results show that the performance of each algorithm is the same as that of the simulation network shown in Section 5.1. It can be seen that the IIT-UP algorithm can minimize the total costs. Fewer name-resolution servers were chosen in level 2 by IIT-DOWN and MDSM. However, the LHP algorithm did not perform well in terms of either the execution time or the total costs.

**Table 3.** Comparison of algorithms on Shanghai Telecom's base stations dataset.

| Algorithm | Execution Time (s) | Level | | | Cost |
|---|---|---|---|---|---|
| | | 3 (0.2 ms) | 2 (0.05 ms) | 1 (0.01 ms) | |
| LHP | 79.49 | 5 | 34 | 302 | 308 |
| Random Allocate (RA) | 101.74 | 6 | 42 | 328 | 342 |
| Minimum Dominator Set in Multilayer (MDSM) | 27.31 | 4 | 27 | 286 | 286 |
| Inter-layer Information Transfer Down (IIT-DOWN) | 28.98 | 4 | 27 | 283 | 283 |
| Inter-layer Information Transfer Up (IIT-UP) | 13.57 | 4 | 37 | 267 | 267 |

## 6. Conclusions

In this paper, we first studied the placement problem of the name-resolution server in multilayer overlay networks and formulated it as an integer linear program problem with the objective of minimizing the deployment costs. Then, two algorithms based on the heuristic ideas of inter-layer information transfer and server reuse were developed. The first was called IIT-DOWN. This algorithm passes the server placement information from the high-level layer to the low-level layer. Servers chosen in the high-level layer are reused to provide low-level services as well. The second one was called IIT-UP, and it follows the same idea but passes information, including both server location and detailed latency information in the opposite direction.

Experiments were conducted on both simulation networks and a real-world dataset. The results demonstrated that our proposed IIT algorithms outperform other existing algorithms in terms of finding more cost-efficient solutions within a shorter execution time. The IIT-DOWN algorithm has the advantage of reducing the number of high-level layer servers. However, IIT-UP can shrink the network scale during its procedure and it greatly reduces the execution. In addition, during the study, we also discovered two facts. One was that the size of the servers grew linearly with the network size in multilayer overlay networks if the physical network was in line with the BA scale-free model. The other one was that average latencies at each layer remained stable while the network size increased.

Future work should focus on two aspects. Firstly, the selection of the set of upper bounds of latencies has a significant influence on the solution of the problem. We only selected the corresponding upper bounds in some typical scenarios, which cannot represent all the scenarios. More upper-bound latency combinations should be evaluated and analyzed to discover the potential rule. Secondly, because name resolution plays an essential role in ICN, having a robust name-resolution service is meaningful. Redundant overlay areas exist during the partition procedure in our proposed algorithms. Further study should aim to make the best of these redundancies to improve the robustness of the name-resolution service.

**Author Contributions:** Conceptualization, J.L., Y.S., and H.D.; methodology, J.L., Y.S.; software, J.L.; writing—original draft preparation, J.L.; writing—review and editing, J.L., Y.S., and H.D.; supervision, Y.S.; project administration, Y.S.; funding acquisition, H.D. All authors have read and agreed to the published version of the manuscript.

**Funding:** This work was funded by Strategic Leadership Project of Chinese Academy of Sciences: SEANET Technology Standardization Research System Development (Project No. XDC02070100).

**Acknowledgments:** We would like to express our gratitude to Jinlin Wang, Yaqin Song, and Luchao Han for their meaningful support for this work.

**Conflicts of Interest:** The authors declare no conflict of interest.

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
