# Peer review of "Two Optimization Algorithms for Name-Resolution Server Placement in Information-Centric Networking"

_applsci, doi:10.3390/app10103588_

Round 1
Reviewer 1 Report
some observations regarding the method applicability in secure name resolution on the Internet may further increase the paper impact to its readersAuthor Response
Please see the attachment.

Reviewer 2 Report
This paper is well written and clear. Novel algorithmic (optimal and heuristic) solutions are proposed by the authors to achieve better results in terms of speed-up w.r.t. state of the art.
Minor editing is needed: please check grammar and resize figures in order to improve readability (some labels are too small).
Reviewer 3 Report
In this study, the authors want to enhance the name resolution in ICN-based network by implementing two algorithms such as IIT-DOWN and IIT-UP. However, there are several comments and suggestions that must be addressed before acceptance.
- According to the current study the title needs to be changed, because it is not clearly related to the area of the study and topic.
- The abstract from line 18 to 26 is too general. It needs to be specific according to the area and topic of the study such as name resolution in ICN or NDN.
- The name resolution is purely related to ICN or ICN. However, the introduction section is consisting of unrelated descriptions and same with abstract. It needs to give reason or should be specific with the topic.
- The study seems as related to ICN-based name resolution system, but the server placement problem is too general. It not defines the current problems for placing the server in ICN.
- The evaluation section is extensively described. However, the simulation scenarios are not clear. Authors have mentioned they have evaluated the proposed work in simulation as well as real world data set there is no any validation of verification of proposed algorithms such as we can show a figure of the scenario in running form. Mostly, for the ICN and NDN NS3 or NDN simulators are broadly used to validate and evaluate the proposed work. Moreover, authors talking about the G5 and IoT technologies in the manuscript, but in the evolution, there is no evaluation scenario is mentioned and it defines as general metric such as latency and cost.
- The evaluation parameters should present in a tabular from to clearly define the evaluation criteria.
- Some references are too old for this topic of interest, cite few of the recently published research articles, such as
- Advancing the State of the Fog Computing to Enable 5G Network Technologies (Sensors, 2020)
- A Comparative Performance Analysis of Popularity-Based Caching Strategies in Named Data Networking (IEEE Access)
